# Vaccine Resistance and Hesitancy among Older Adults Who Live Alone or Only with an Older Partner in Community in the Early Stage of the Fifth Wave of COVID-19 in Hong Kong

**DOI:** 10.3390/vaccines10071118

**Published:** 2022-07-13

**Authors:** Dexing Zhang, Weiju Zhou, Paul Kwok-Ming Poon, Kin On Kwok, Tracy Wai-Sze Chui, Phoebe Hoi Yi Hung, Bonny Yin Tung Ting, Dicken Cheong-Chun Chan, Samuel Yeung-Shan Wong

**Affiliations:** 1JC School of Public Health and Primary Care, The Chinese University of Hong Kong, Hong Kong, China; zhangdxdaisy@cuhk.edu.hk (D.Z.); joewjzhou@cuhk.edu.hk (W.Z.); kwokmingpoon@cuhk.edu.hk (P.K.-M.P.); kkokwok@gmail.com (K.O.K.); waiszetracychui@cuhk.edu.hk (T.W.-S.C.); phoebehung@cuhk.edu.hk (P.H.Y.H.); bonnyting@cuhk.edu.hk (B.Y.T.T.); dicken@cuhk.edu.hk (D.C.-C.C.); 2Stanley Ho Centre for Emerging Infectious Diseases, The Chinese University of Hong Kong, Hong Kong, China; 3Hong Kong Institute of Asia-Pacific Studies, The Chinese University of Hong Kong, Hong Kong, China; 4Shenzhen Research Institute, The Chinese University of Hong Kong, Shenzhen 518172, China

**Keywords:** COVID-19, vaccination, vaccine resistance and hesitancy, older people, living alone, living with an older partner

## Abstract

Vaccination is an effective way in providing protection against COVID-19 infection and severe outcomes. However, vaccine resistance and hesitancy are a great concern among vulnerable populations including older adults who live alone or only with an older partner. This study examined their vaccination status and reasons and associated factors of vaccine resistance and hesitancy. A cross-sectional study was conducted among older adults living alone or only with an older partner in communities in Hong Kong. Participants were interviewed between October 2021 and February 2022. Logistic regression analyses were employed to examine factors associated with vaccine resistance and hesitancy. Of the 2109 included participants, the mean age was 79.3 years (SD 7.6), 1460 (69.2%) were female, 1334 (63.3%) lived alone, and 1621 (76.9%) were receiving social security support. The vaccine uptake, non-uptake (i.e., resistance), and hesitancy rates were 50.1%, 34.4%, and 15.5%, respectively. The top four reasons for vaccine resistance and hesitancy were “Not feeling in good health” (27%), “Worry about vaccine side effects” (18%), “Feeling no need” (10%), and “Lack of recommendation from doctors” (9%). Vaccine resistance and hesitancy was significantly associated with older age, living alone, more chronic conditions, fewer types of social media use, and lower self-rated health status. Similar associations can be observed in their separate analysis for vaccine resistance and vaccine hesitancy, and ever hospital admission over the past 6 months was additionally related to vaccine hesitancy. Older people who live alone or only with an older partner had a low vaccination rate. Poor health or worry about vaccine side effects were the most common reasons for their vaccine resistance and hesitancy. Actions are greatly needed to improve the uptake rate among this vulnerable population, especially those who were older, have poorer health, and use less social media.

## 1. Introduction

Coronavirus disease 2019 (COVID-19) is a respiratory disease that causes symptoms such as fever, dry cough, and shortness of breath [1]. Since first reported 2.5 years ago, more than 526 million people around the world have been infected by COVID-19 and more than 6 million people have died of the disease [2]. Mass vaccination of populations is considered to be the most effective way to prevent and control the COVID-19 pandemic worldwide. Vaccinating older adults against COVID-19 is currently one of the public health priorities across the world [3]. However, the current literature predominately examines the vaccination acceptance in the general population [4] or levels of knowledge concerning COVID-19 in older adults [5]. Studies that have determined the resistance and hesitancy of the older population to be vaccinated are relatively limited [3]. In contrast to those in other countries or regions with high vaccine acceptance rates, such as the UK (98.7%) [6], mainland China (90.9%) [7], and Taiwan (74.9%) [8], in Hong Kong many older adults who live in the community are resistant and hesitant to receive the COVID-19 vaccine [9], where a low vaccination take-up rate among older adults due to vaccine resistance or hesitancy is still the major challenge to achieve high levels of vaccination coverage.

Vaccine hesitancy refers to a delay in acceptance or refusal of vaccination even though the vaccination services are available [10]. Despite COVID-19 vaccines being readily available since February 2021, Hong Kong residents had the most vaccine hesitancy in Asia and the Pacific [11], and the age group with the highest vaccine hesitancy has changed from young adults to older adults aged ≥65 years after the implementation of the vaccination programme [9]. The vaccine resistance and hesitancy, widespread among the older population, have led Hong Kong to the highest daily death rate per capita in the world [12]. There were about 9000 coronavirus-related deaths in the fifth COVID-19 epidemic wave as of late April 2022, where more than 70% of them were aged 80 or older and 73% of whom were unvaccinated [13].

Numerous studies have been conducted to identify factors associated with vaccine resistance and hesitancy among different subpopulations in Hong Kong, including working-age people [14] and school students [15], but few studies have been conducted among older adults, particularly those who live alone or only with an older partner, an “at-risk” group who are often socially isolated, have poor physical and mental health, and have unmet care needs [16]. Hong Kong is an aging society where the proportion of the population aged 65 or above has increased to nearly 20% in 2021 [17]. Understanding factors associated with vaccine resistance and hesitancy among older adults is essential for the policymakers to refine current strategies to increase the vaccine uptake rate of this vulnerable group. Therefore, in this study, we examined the uptake rate of COVID-19 vaccines among older adults who live alone or only with an older partner in Hong Kong, and to identify reasons and potential factors associated with vaccine resistance and hesitancy. We hypothesized that health status and vaccine side effects might be the main reasons why they were resistant and hesitant to get vaccinated, and those who were more ill (e.g., with more chronic conditions and poorer self-rated health) were more likely to have vaccine resistance and hesitancy.

## 2. Methods

A community-based cross-sectional survey was conducted among older adults who live alone or only with an older partner in Hong Kong. Participants were recruited from their existing members and community social networks, and assessed by seven collaborating non-governmental organizations (NGOs), which provided services to older residents across different districts in Hong Kong between 20 October 2021 and 08 February 2022, during which time, Hong Kong started by far the most severe wave (i.e., the 5th wave) of COVID-19 infection since December 2021. Informed consent was obtained from all participants before they could proceed to the survey items. This survey was approved by The Joint Chinese University of Hong Kong–New Territories East Cluster Clinical Research Ethics Committee (Reference number: 2020.487).

### 2.1. Participants and Data Collection 

The participants were eligible to be included if they were (1) aged 60 or above, (2) living alone or only with an older partner, and (3) not institutionalized (i.e., currently living in the community). No specific exclusion criteria were adopted. However, those older adults with severe communication difficulties (e.g., hearing problems) were less likely to be included. During the interview, either telephone or face-to-face interviews were conducted by trained NGOs’ social workers or volunteers. The survey data were collected by either using paper-based or web-based questionnaires and managed using RedCap (Research Electronic Data Capture) [18], which is a secure and web-based data capture tool with a local host at The Chinese University of Hong Kong.

All collected data were entered into the RedCap platform by trained community social workers and volunteers. The entered data were checked by a trained research staff case by case and then randomly checked by another researcher to ensure data quality. Any missing or unreasonable data were then highlighted in a password encrypted file, which was then double checked and improved by the NGO workers. A total of 2109 completed and valid responses were analyzed.

### 2.2. Measures

The questionnaire was developed based on our previous studies involving older adults and vaccination status [19,20]. The survey included some non-standard questions, which were developed by a group of epidemiologists, doctors, nurses, and social workers. To further improve the clarity and suitableness of our survey questions and answers, the questionnaire was pilot tested in several elderly centers among more than 30 older adults, and minor revisions were further made. In the current study, the reliability was evaluated by Cronbach’s alpha (internal consistency coefficient), with an alpha value of 0.729, showing an acceptable level of reliability.

Vaccination status was measured by a question, “Have you ever been vaccinated against COVID-19?”. The answer options included four categories: (1) “Have been vaccinated”, (2) “Will be vaccinated in the coming 12 months”, (3) “Have not decided to be vaccinated”, and (4) “Will not be vaccinated”. The participants were further asked the reasons why they were willing or unwilling to be vaccinated. Those who answered (3) or (4) were regarded as having vaccine hesitancy and resistance, respectively.

Other items in the survey included demographics (age, gender, living status, occupational status, socioeconomic status (SES)), social networks and support (social media use, perceived help available when needed), medical information (hospital admission in the past 6 months, medical appointment in the next 6 months), health behaviors (regular measurements of blood pressure and blood glucose), physical and mental health (doctor-diagnosed chronic diseases, loneliness, depression, anxiety), and subjective wellbeing (subjective memory decline, self-rated health status, perceived meaning in life). Depression and anxiety symptoms were assessed by the two-item Patient Health Questionnaire (PHQ-2) [21] and the two-item Generalized Anxiety Disorder Questionnaire (GAD-2) [22], with scores of ≥3 indicating positive and higher scores denoting higher severity. The Chinese validated three-item UCLA Loneliness Scale (UCLA-3) was used to measure loneliness (range 3–9), with higher scores representing more serious loneliness [23]. A question on self-rated overall health was measured with a score between 0 (unhealthy) and 100 (healthy). Perceived meaning in life was assessed using one item extracted from the validated reliable Chinese Purpose in Life test (CPIL) [24]: “My personal existence is utterly meaningless and without purpose vs. very purposeful and meaningful”. Participants were asked to rate their perceived meaning of existence from a 7-point Likert scale, with 1 denoting the lowest level and 7 the highest level.

### 2.3. Statistical Methods

Data were described with numbers, percentages, means, and standard deviations (SD). In the current study, missing data that occurred in covariates were treated as a special group for analysis since those participants who missed information on a variable may have a special reason for not answering the survey question.

We employed univariate logistic regression models to estimate odds ratios (ORs) and their 95% confidence intervals (CIs) to identify potential factors associated with vaccine resistance and hesitancy. A multivariate regression model, which included all variables with *p* < 0.1 in the univariate analysis or potentially important factors (e.g., gender, SES), was used to further explore their independent associations with the vaccination decision. We also analyzed data on vaccine resistance and vaccine hesitancy separately to identify their potential related factors. As it would need 15 samples for one variable for the regression model [25], the minimum required sample size would be 300 with about 20 variables in the models. All analyses were performed using the SPSS statistical package (Windows version 26.0; SPSS Inc., Chicago, IL, USA).

## 3. Results

### 3.1. Participants’ Characteristics and Reasons for Vaccine Resistance or Hesitancy

Of the 2109 participants, the mean age was 79.3 years (SD 7.6), 1460 (69.2%) were female, 1334 (63.3%) were living alone, and 1621 (76.9%) were receiving social security support (Comprehensive Social Security Assistance (CSSA) or Old Age Living Allowance (OALA)). Among all eligible participants, 920 (43.6%) reported they had received the COVID-19 vaccine, 137 (6.5%) planned to be vaccinated in the next 12 months, 725 (34.4%) would not get a vaccination, and 327 (15.5%) were hesitant towards vaccination. The vaccine uptake rates were much lower for participants aged 80 or above (35.8%) compared with those aged 60–69 (52.0%) and 70–79 years (50.5%). The main reasons of vaccine resistance or hesitancy were “Not feeling in good health” (N = 574, 27.2%), “Worry about vaccine side effects causing complications” (N = 389, 18.4%), “Feeling no need” (N = 201, 9.5%), and “Lack of recommendation from doctors” (N = 191, 9.1%) (Figure 1). Table 1 presents the distribution of participants’ characteristics by vaccination status.

### 3.2. Univariate Analysis on Factors Associated with Vaccine Resistance and Hesitancy

Univariate analysis suggested that older age, living alone, retirement status, more chronic conditions, ever hospital admission over the past 6 months, perceived no help available when needed, fewer types of social media use, lower levels of self-rated health status and meaning in life, and higher levels of loneliness, memory loss, depression, and anxiety were significantly associated with vaccine resistance and hesitancy (Table 2).

### 3.3. Multivariate Analysis on Factors Associated with Vaccine Resistance and Hesitancy

In the multivariate analysis (Table 2), we still observed significant associations of vaccine resistance and hesitancy with older age (aOR (95% CI): 1.03 (1.02–1.04), *p* < 0.001), living alone (aOR (95%CI): 1.32 (1.08–1.61), *p* = 0.007), more chronic conditions (compared to 0–1 chronic condition, aOR (95%CI) was 1.55 (1.17–2.04) for > 3 chronic conditions, *p* = 0.002,) fewer types of social media use (comparing to no social medial use in the past 2 weeks, aOR (95%CI): 0.69 (0.55–0.86) for 1 type use, *p* = 0.001; 0.48 (0.36–0.65) for > 1 types use, *p* < 0.001), and lower level of self-rated health (comparing to high score (67–100), aOR (95%CI) was 1.51 (1.20–1.91) for middle score (34–66), *p* < 0.001; 1.57 (1.24–1.99) for low score (0–33), *p* < 0.001).

**Table 2 vaccines-10-01118-t002:** Univariate (crude OR) and multivariate (adjusted OR) analyses on factors associated with COVID-19 vaccine reluctance and hesitancy to be vaccinated (N = 2109) (95% CI: 95% confidence intervals).

Characteristics	Crude OR	95% CI		*p*-Value	Adjusted OR *	95 % CI		*p*-Value
**Age (years)**								
Mean (SD)	1.04	1.03	1.05	**<0.001**	1.03	1.02	1.04	**<0.001**
**Sex**								
Male	Ref				Ref			
Female	0.94	0.78	1.13	0.493	0.91	0.74	1.11	0.343
**Living status**								
Live with an older partner	Ref				Ref			
Live alone	1.38	1.16	1.65	**<0.001**	1.32	1.08	1.61	**0.007**
**Occupational status**								
Retirement	Ref				Ref			
Caring family/full-time/part-time/unemployed	0.47	0.30	0.74	**0.001**	0.63	0.39	1.02	0.062
**Social security support**								
No	Ref				Ref			
Yes (CSSA or OALA)	1.22	0.99	1.49	0.057	1.00	0.81	1.25	0.978
**Number of doctor-diagnosed chronic conditions**								
0–1	Ref				Ref			
2–3	1.17	0.97	1.42	0.110	1.03	0.84	1.28	0.764
>3	1.89	1.49	2.39	**<0.001**	1.55	1.17	2.04	**0.002**
**Ever hospital admission over the past 6 months**								
No	Ref				Ref			
Yes	1.51	1.18	1.93	**0.001**	1.24	0.95	1.60	0.114
**Medical appointment in the next 6 months**								
No	Ref							
Yes	1.14	0.91	1.42	0.265				
**Regular measurement of blood pressure**								
No hypertension	Ref							
Yes	1.13	0.85	1.50	0.396				
Not measure regularly	1.28	0.93	1.74	0.126				
**Regular measurement of blood glucose**								
No diabetes	Ref				Ref			
Yes	1.21	0.97	1.51	0.091	1.08	0.85	1.39	0.521
Not measure regularly	1.17	0.88	1.55	0.294	0.98	0.71	1.33	0.879
**Help available when needed**								
Yes	Ref				Ref			
No	1.24	1.01	1.53	**0.038**	1.13	0.90	1.42	0.279
**Number of social media use in the past 2 weeks**								
0	Ref				Ref			
1	0.62	0.50	0.76	**<0.001**	0.69	0.55	0.86	**0.001**
>1	0.39	0.29	0.52	**<0.001**	0.48	0.36	0.65	**<0.001**
**Self-rated Health status in score**								
High (67–100)	Ref				Ref			
Middle (34–66)	1.42	1.14	1.76	**0.002**	1.51	1.20	1.91	**<0.001**
Low (0–33)	1.76	1.43	2.17	**<0.001**	1.57	1.24	1.99	**<0.001**
**Meaning of life**								
High (5–7)	Ref				Ref			
Middle (3–4)	1.41	1.16	1.72	**0.001**	1.10	0.89	1.37	0.391
Low (1–2)	1.62	1.02	2.57	**0.039**	1.08	0.65	1.79	0.771
Missing	1.43	1.11	1.84	**0.005**	1.34	1.02	1.77	**0.036**
**Loneliness (measured by UCLA-3)**								
0–5	Ref				Ref			
≥6	1.44	1.14	1.81	**0.002**	1.11	0.85	1.46	0.433
**Memory loss**								
No	Ref				Ref			
Yes, but not worry	1.02	0.84	1.25	0.817	0.91	0.73	1.12	0.375
Yes, and worry	1.44	1.10	1.89	**0.009**	1.08	0.80	1.46	0.620
**Depression (measured by PHQ-2)**								
0–2	Ref				Ref			
≥3	1.58	1.13	2.20	**0.007**	1.07	0.66	1.75	0.785
**Anxiety (measured by GAD-2)**								
0–2	Ref				Ref			
≥3	1.61	1.11	2.33	**0.012**	1.27	0.74	2.19	0.380

* Multivariate regression (only variables with *p* < 0.1 in univariate analysis were included): adjusted for age, sex, living status, occupational status, social security support, number of chronic conditions, hospital admission, regular measurement of blood glucose, help available when needed, number of social medial use, self-rated health status, meaning of life, loneliness, memory loss, depression and anxiety.

### 3.4. Additional Separate Analysis on Factors Associated with Vaccine Resistance and Factors Associated with Vaccine Hesitancy

The individual analysis for factors associated with vaccine resistance and factors associated with vaccine hesitancy showed similar patterns of ORs for these factors to those in their combined analysis (Appendix A). Vaccine resistance and vaccine hesitancy were significantly associated with older age, living alone, more chronic conditions, fewer types of social media use, and lower level of self-rated health (Appendix A), and vaccine hesitancy was additionally associated with ever hospital admission over the past 6 months (Appendix A).

## 4. Discussion

### 4.1. Major Findings 

The study reported the overall COVID-19 vaccine uptake rate among older people who lived alone or only with an older partner in Hong Kong. Participants aged 80 or above had much lower vaccine uptake rate than those aged 60–69 and 70–79 years. The main reasons for vaccine resistance or hesitancy were poor health and worry about vaccine side effects causing complications. Despite the fact that older people with poor health (e.g., more chronic conditions) urgently need vaccination, multivariate regression analysis suggested that vaccine resistance and hesitancy were significantly associated with older age, having more chronic conditions, having a lower level of self-rated health status, living alone, and using fewer types of social media. In addition, older adults who had past hospitalization were hesitant to get vaccinated.

Despite repeated urgent appeals by authorities and medical experts for older adults to take a vaccine, our results showed that COVID-19 vaccination coverage among older adults who lived alone or only with an older partner was low with only half of the participants (50.1%) having already received the vaccines. It further showed that more than 34% and 15% reported that they would definitely or uncertainly not get vaccinated. Consistent with previous qualitative studies [26,27], “Not feeling in good health” and “Worry about vaccine side effects causing complications” were considered important reasons by 27% and 18% of older participants, respectively, for refusing or delaying receiving COVID-19 vaccines. In our multivariate analysis, both subjective and objective health status, i.e., self-rated health status and number of chronic conditions (or ever hospital admission), respectively, are significantly associated with vaccine resistance and hesitancy among older people, which is consistent with a recent study conducted in Shanghai, an epicenter of the COVID-19 outbreak in China [7]. Around the time that the COVID-19 vaccines were first rolling out, the widespread misinformation on the side-effects of vaccines such as frequent heart attacks and severe allergies negatively impacted vaccination intent among older people with multimorbidity [28]; although, a recent Hong Kong study using population-based vaccination records suggested that vaccinated patients have lower risks of adverse events than unvaccinated individuals [29]. In addition, doctors’ recommendation is one of the factors for older adults in vaccine decision making since they were unclear whether they were appropriate to take vaccines and would like to ask their doctors. Nevertheless, in our previous study, over 35% of frontline doctors would not recommend their patients without contraindications for the vaccination in practice [20], which could be due to the uncertainties on benefits and side effects when vaccines were first available. These might be the reasons why the non-vaccination rate was high among older adults. Apart from increasing knowledge about vaccine efficacy, efforts should be made by healthcare workers such as physicians and medical scientists to promote vaccine safety among older adults with multimorbidity to boost the vaccination rate.

Our study showed that older people who lived alone were less likely to get vaccinated than those who lived with their older partners. The “hidden elderly”—those without family support and those who are disengaged from the community and disadvantaged, yet not helped by available services and support—are more vulnerable to increased isolation during the pandemic [30]. Living alone as an overlooked factor in vaccination programs needs to be considered; a 2017 systematic review reported that older adults living alone had a 39–71% lower vaccine uptake rate than those not living alone [31]. This might be because older people living alone might perceive that there is no need in being vaccinated in order to protect their older partner at home or they might be worried about what will happen if they develop complications after getting vaccinated and will be left helpless [26]. More community support and assistance, i.e., checking on these residents through home visits or phone calls and following up after they have been vaccinated, are needed by the governmental and non-governmental organizations (e.g., community elderly centers). 

Our study found that almost two-thirds of older people who lived alone or only with an older partner did not use social media. It was further found that fewer types of social media use were associated with vaccine resistance and hesitancy. On one hand, updated vaccine information and knowledge (e.g., benefits and side effects) is less likely to reach older adults who do not use social media since this information is more likely to be easily seen and searched on the Internet than on other conventional media channels. On the other hand, older people may have potential barriers to accessing the COVID-19 vaccine since the vaccine is predominantly scheduled online. Older adults may lack Internet access, be unfamiliar with online scheduling, and lack a social network that might assist in traveling to vaccination sites. It would then need effective measures to have information on vaccine protection, safety, and services reach these older adults, e.g., education and providing walk-in or escort services for them to easily have the vaccine.

Our study also found that almost 10% of participants reported that they felt no need to take the vaccine. Over the past two years, Hong Kong has adopted the dynamic zero-COVID strategy, which may foster vaccine hesitancy among older adults due to a lack of urgency to get vaccinated [32,33], particularly when there was widespread misinformation on the side-effects in relation to the COVID-19 vaccines [28]. Even under the new vaccine pass policy that people need to have three doses to access many public and recreational places unless exempted, the vaccination rate in older people aged 80+ was 67% on 7 June 2022 [34], much less than in other countries such as Singapore [35]. Nevertheless, stepping up vaccinations among older people through promoting communication on their effectiveness and safety should be given priority by the healthcare workers and policy makers.

### 4.2. Strengths and Limitations

This study investigated a vulnerable population—older adults who live alone or only with an older partner in the community needing more support and attention during COVID-19. To our best knowledge, it is the first study to examine vaccination rate, reasons for vaccine resistance and hesitancy, and their related factors among this population. The study also covered a relatively large sample with reliable findings in the statistical models, with various factors also having been adjusted for. The findings have filled some knowledge gaps about vaccine resistance and hesitancy among vulnerable populations. The results will help policy makers, practitioners, and researchers develop strategies targeting this vulnerable group to improve vaccination rates.

The study had several limitations. First, this study was conducted during the early stage of the fifth wave of COVID-19 in Hong Kong, a period during which social distancing was encouraged by the authorities. The distribution of the questionnaire depended on the elderly centers’ connections; it was therefore a convenience sample rather than a random one. The study might not have included a representative sample in the community; the participants were far more likely to be of older age, be of female sex, have lower SES, and have lower COVID-19 vaccine uptake rates compared to the general older population in Hong Kong [34,36,37]. Thus, the generalizability of our findings may be limited. However, due to the collaborating elderly community centers covering 13 out of 18 different districts with different socio-economic status in Hong Kong, the results could still provide important insights into real situations. More studies with a random sample of older adults (e.g., household surveys) who live alone or only with an older partner are required to examine the factors associated with vaccine resistance and hesitancy. Second, most variables have some missing data. The largest number missing is in the variable of meaning of life (N = 347, 15.0%), since this question was to be answered optionally, depending on participants’ emotions during the interview. Participants who did not answer this question were more likely to have more chronic conditions, hospital admissions over the past 6 months, medical appointments in the next 6 months, a need to measure blood pressure, and fewer social medial use (Appendix A), which all were associated with increased odds of vaccine resistance and vaccine hesitancy. This might be one of the reasons for the strongest associations of the category of “missing data” in meaning of life with vaccine resistance and hesitancy after adjustment for all covariates. Further studies are needed to investigate the reasons why participants were not willing to answer this question.

### 4.3. Implications

Our findings highlighted the low vaccination rate among older adults who live alone or only with an older partner. Concerns about health status and vaccine side effects were the main reasons why older people were resistant and hesitant to be vaccinated. Future health education/promotion on only limited contradictions and little side effects of vaccines are important to achieve high levels of vaccination coverage in this population. It is vital to connect with community health service centers to share clear and timely information about COVID-19 vaccines (e.g., safety and efficacy) and vaccination sites and address misinformation. Healthcare professionals and workers may be encouraged to help answer inquiries from the elderly to strengthen their confidence in vaccination. Tailored messages and content, both online and offline (e.g., leaflets), which are relative to older adults, are required to increase their engagement. Follow-up services (e.g., phone calls) after getting vaccinated would also help relieve their vaccine anxiety.

## 5. Conclusions

The overall COVID-19 vaccine uptake rate was much lower among older people who lived alone or only with an older partner than in the general public in Hong Kong. The most common reasons for their vaccine resistance or hesitancy were due to ill health or worry about vaccine side effects or complications after vaccination. Older people who were living alone had more chronic conditions and had less social media use, and were more likely to have vaccine resistance and hesitancy. Strategies and actions, through promoting the safety and effectiveness of the COVID-19 vaccines, are urgently needed to improve the vaccine uptake rate among this vulnerable population.

## Figures and Tables

**Figure 1 vaccines-10-01118-f001:**
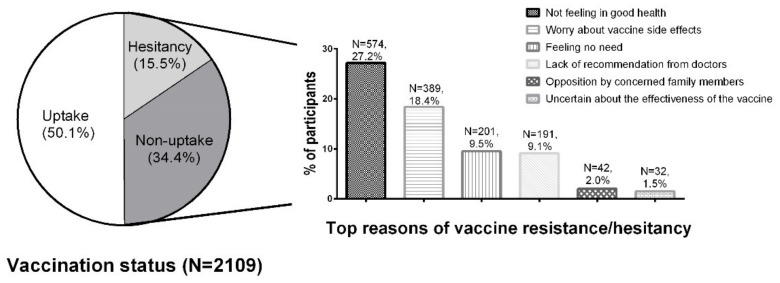
Vaccination status and top reasons of COVID-19 vaccine resistance or hesitancy among older participants who lived alone or only with an older partner.

**Table 1 vaccines-10-01118-t001:** Characteristics and vaccination status among older people who live alone or only with an older partner in Hong Kong.

Variables	All Participants		Vaccinated or Intention to Be Vaccinated *		Hesitated to Be Vaccinated		Unvaccinated	
	N = 2109	%	N = 1057	%	N = 327	%	N = 725	%
**Age (years)**								
Mean (SD)	79.3	7.6	78.2	7.2	79.7	7.5	80.8	8.0
**Sex**								
Male	649	30.8	318	30.1	91	27.8	240	33.1
Female	1460	69.2	739	69.9	236	72.2	485	66.9
**Living status**								
Living with an older partner	775	36.7	428	40.5	111	33.9	236	32.6
Living alone	1334	63.3	629	59.5	216	66.1	489	67.4
**Occupational status**								
Retirement	2007	95.2	990	93.7	316	96.6	701	96.7
Caring family/Full-time/part-time/unemployed	92	4.4	62	5.9	9	2.8	21	2.9
Missing	10	0.5	5	0.5	2	0.6	3	0.4
**Social security support**								
No	488	23.1	263	24.9	74	22.6	151	20.8
Yes (CSSA or OALA)	1621	76.9	794	75.1	253	77.4	574	79.2
**Number of doctor-diagnosed chronic conditions**								
0–1	749	35.5	414	39.2	105	32.1	230	31.7
2–3	904	42.9	464	43.9	148	45.3	292	40.3
>3	447	21.2	177	16.7	74	22.6	196	27.0
Missing	9	0.4	2	0.2	0	0.0	7	1.0
**Ever hospital admission over the past 6 months**								
No	1787	84.7	925	87.5	266	81.3	596	82.2
Yes	306	14.5	127	12.0	59	18.0	120	16.6
Missing	16	0.8	5	0.5	2	0.6	9	1.2
**Medical appointment in the next 6 months**								
No	375	17.8	198	18.7	46	14.1	131	18.1
Yes	1721	81.6	854	80.8	280	85.6	587	81.0
Missing	13	0.6	5	0.5	1	0.3	7	1.0
**Regular measurement of blood pressure**								
No hypertension	226	10.7	121	11.4	25	7.6	80	11.0
Yes	1335	63.3	674	63.8	215	65.7	446	61.5
Not measure regularly	533	25.3	253	23.9	86	26.3	194	26.8
Missing	15	0.7	9	0.9	1	0.3	5	0.7
**Regular measurement of blood glucose**								
No diabetes	1452	68.8	748	70.8	236	72.2	468	64.6
Yes	398	18.9	186	17.6	60	18.3	152	21.0
Not measure regularly	216	10.2	103	9.7	23	7.0	90	12.4
Missing	43	2.0	20	1.9	8	2.4	15	2.1
**Help available when needed**								
Yes	1623	77.0	831	78.6	254	77.7	538	74.2
No	472	22.4	216	20.4	71	21.7	185	25.5
Missing	14	0.7	10	0.9	2	0.6	2	0.3
**Types of social media use in the past 2 weeks**								
0	1339	63.5	595	56.3	220	67.3	524	72.3
1	506	24.0	286	27.1	74	22.6	146	20.1
>1	253	12.0	170	16.1	32	9.8	51	7.0
Missing	11	0.5	6	0.6	1	0.3	4	0.6
**Self-rated Health status**								
High (score 67–100)	686	32.5	397	37.6	88	26.9	201	27.7
Middle (score 34–66)	624	29.6	307	29.0	113	34.6	204	28.1
Low (score 0–33)	760	36.0	333	31.5	121	37.0	306	42.2
Missing	39	1.8	20	1.9	5	1.5	14	1.9
**Meaning of life**								
High (5–7)	1075	51.0	586	55.4	166	50.8	323	44.6
Middle (3–4)	638	30.3	293	27.7	90	27.5	255	35.2
Low (1–2)	80	3.8	34	3.2	12	3.7	34	4.7
Missing	316	15.0	144	13.6	59	18.0	113	15.6
**Loneliness (measured by UCLA-3)**								
0–5	1748	82.9	902	85.3	268	82.0	578	79.7
≥6	355	16.8	151	14.3	59	18.0	145	20.0
Missing	6	0.3	4	0.4	0	0.0	2	0.3
**Memory loss**								
No	560	26.6	291	27.5	78	23.9	191	26.3
Yes, but not worry	1201	56.9	617	58.4	187	57.2	397	54.8
Yes, and worry	340	16.1	146	13.8	61	18.7	133	18.3
Missing	8	0.4	3	0.3%	1	0.3	4	0.6
**Depression (measured by PHQ-2)**								
0–2	1929	91.5	984	93.1	307	93.9	638	88.0
≥3	156	7.4	62	5.9	18	5.5	76	10.5
Missing	24	1.1	11	1.0	2	0.6	11	1.5
**Anxiety (measured by GAD-2)**								
0–2	1957	92.8	996	94.2	307	93.9	654	90.2
≥3	125	5.9	49	4.6	16	4.9	60	8.3
Missing	27	1.3	12	1.1	4	1.2	11	1.5

SD = standard deviation, PHQ = Patient Health Questionnaire, GAD = Generalized Anxiety Disorder Scale, CSSA = Compressive Social Security Scheme, OALA = Old Age Living Allowance. * Among participants in this subgroup, 137 participants responded to the survey item “I will take COVID-19 vaccines in the next 12 months”.

## Data Availability

The data presented in this study are available on request from the corresponding author (yeungshanwong@cuhk.edu.hk). The data are not publicly available as this is an ongoing study.

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
