# Peer review of "Vaccine Resistance and Hesitancy among Older Adults Who Live Alone or Only with an Older Partner in Community in the Early Stage of the Fifth Wave of COVID-19 in Hong Kong"

_vaccines, 2022, doi:10.3390/vaccines10071118_

Round 1

Reviewer 1 Report

The author need to mention clearly in the introduction about the situation in other countries not only Hong Kong.

Explain clearly the research hypothesis and the rational of the study.

More information are needed for participants inclusion criteria in the method section

In discussion, the authors need to discuss the results of table 3 too

Author Response

Response:

Thank the reviewer for the valuable advice. We have added more about the situation in other countries or regions accordingly in the revised manuscript, Page 4, Lines 10-15 “In contrast to those in other countries or regions with high vaccine acceptance rates such as UK (98.7%) , mainland China (90.9%) , and Taiwan (74.9%) , in Hong Kong many older adults who live in the community are resistant and hesitant to receive the COVID-19 vaccine , where a low vaccination take-up rate among older adults due to vaccine resistance or hesitancy is still the major challenges to achieve high levels of vaccination coverage”.

Explain clearly the research hypothesis and the rational of the study.

Response:

Thank you very much for your comments. During the 5th wave of COVID-19 outbreak, Hong Kong has the highest daily death rate per capita in the world. The vaccine resistance and hesitancy widespread among older population could be one of the main reasons. In terms of coronavirus related deaths, 70% of them were aged 80 or older and 73% of whom were unvaccinated. Improving vaccination rate among older adults is urgent. Studies understanding reasons and factors associated with their vaccination resistance and hesitancy could provide insights for effective strategies to achieve high levels of vaccination coverage. This study tried to identify reasons and potential factors associated with vaccine resistance and hesitancy among a vulnerable group – older adults who live alone or only with an older partner in Hong Kong. The results would fill some of the knowledge and information gap.

We have included research hypothesis and rational of the study accordingly, Page 5, Lines 9-17, “Understanding factors associated with vaccine resistance and hesitancy among older adults is essential for the policymakers to refine the current strategies to increase vaccine uptake rate of this vulnerable group”;

We hypothesized that health status and vaccine side effects might be the main reasons they were resistant and hesitate to get vaccinated, and those who were more ill (e.g., with more chronic conditions and poorer self-rated health) were more likely to have vaccine hesitancy and resistance.”

More information is needed for participants inclusion criteria in the method section

Response:

Thank you, as mentioned in the manuscript, Page 6, Lines 8-9, the eligibility criteria were being aged 60 years or above, living alone or only with an older partner, and currently living in the community.  We have updated slightly with further information: The participants were eligible to be included if they were (1) aged 60 or above, (2) living alone or only with an older partner, and (3) not institutionalized (i.e., currently living in the community). No specific exclusion criteria were adopted. However, those older adults with severe communication difficulties (e.g., hearing problems) were less likely to be included.”

In discussion, the authors need to discuss the results of table 3 too

Response:

Thanks for your comments. Results of Table 3 have been discussed. In our Discussion section, factors that were significantly associated with vaccine resistance and hesitancy have been compared with previous studies and discussed further, including older age, living alone, number of chronic conditions, types of social media use, and self-rated health status, Pages 10-12. In addition, we have added more practical recommendations according to the results of Table 3 in the Discussion section, Page14, Lines 14-23, and Page 15, Lines 1-2.

Our findings highlighted the low vaccination rate among older adults who lived alone or only with an older partner. Concerns on health status and vaccine side effects were the main reasons why older people were resistant and hesitant to be vaccinated. Future health education/promotion on limited contradictions and little side effects of vaccine are important to achieve high levels of vaccination coverage in this population. It is vital to connect with community health service centres to sharing clear and timely information about COVID-19 vaccines (e.g., safety and efficacy) and vaccination sites and addressing misinformation. Healthcare professionals and workers may be encouraged to help answer enquiries from the elderly to strengthen their confidence in vaccination. Tailored messages and content, both online and offline (e.g., leaflets), that are relative to older adults are required to increase their engagement. Follow-up services (e.g., phone calls) after getting vaccinated would also help relieve their vaccine anxiety.”    

Reviewer 2 Report

First of all I would like to thank for the opportunity to review this paper. COVID-19 is an ongoing pandemic that has resulted in global health, economic and social crises. Actually, the vaccination campaign is the first method to counteract the COVID-19 pandemic; however, sufficient vaccination coverage is conditioned by the people’s acceptance of these vaccines, especially in high risk population. In this context, the paper under review is aimed at evaluating the vaccination status and the reasons and associated factors of vaccine resistance and hesitancy in a sample of older adults who live alone or only with an older partner.

The subject under study is certainly important, especially in the historical period we are experiencing. The article presents interesting results but, but it is nevertheless believed that, given the organization of the contents and the description of the same, the manuscript cannot be published in its current form, especially for the local impact. I would like to encourage authors to consider several issues to be improved.

Title: it can be improved, highlight the object of the study: place, time and person.

Introduction: The authors should make clearer what is the gap in the literature that is filled with this study. The authors must better frame their study within the vast body of literature that addressed the issue of acceptance of the vaccination in the adult population and their level of knowledge concerning COVID-19 in older adults (refer to articles with DOI: https://doi.org/10.3390/ijerph182010872).

Methods: The survey was conducted, in part, using non-standard questions. The use of an unreliable instrument is a serious and irreversible limitation. The fact that a similar questions have been used in previous surveys is not sufficient. A validation process must be performed to evaluate the added questions to standard questionnaires. At least an evaluation of the reliability must be performed. Was a pilot study performed?

The enrolment procedure must be specified. How did the authors choose the way to select the sample? And how to perform a direct interview or an online questionnaire. This can represent a great bias origin. How did they avoid the selection bias? The author do not report a minimum sample size. What is the reference population? How large is it? Without the numerical identification of the reference population is not clear the validity of the study. A non-representative sample is by its self a non-sense-survey.

Statistical analysis: I suggest to insert a measure of the magnitude of the effect for the comparisons. Please consider to include effect sizes.

Discussion: I also suggest expanding. Emphasize the contribution of the study to the literature. The discussion must be updated with the comparison and discussion regarding knowledge in this group of population (see the above mentioned reference). The Authors should add more practical recommendations for the reader, based on their findings. Also, the section of limitations and future search is also very short, the Authors could elaborate on that.

Author Response

Response:

We appreciate the reviewer for the positive and very valuable comments on our paper. Vaccinating older adults, particularly those who live alone or only with an older partner, is one of the public health priorities across the world. Understanding reasons why older adults were resistant and hesitant and associated factors would be of significance to achieve high levels of vaccination coverage. We have revised and improved our manuscripts accordingly. The one-to-one responses are shown below.

Title: it can be improved, highlight the object of the study: place, time and person.

Response:

Thank you. We have revised the title accordingly as “Vaccine resistance and hesitancy among older adults who live alone or only with an older partner in community in the early stage of the fifth wave of COVID-19 in Hong Kong”, Page 1, Lines 1-3.

Introduction: The authors should make clearer what is the gap in the literature that is filled with this study. The authors must better frame their study within the vast body of literature that addressed the issue of acceptance of the vaccination in the adult population and their level of knowledge concerning COVID-19 in older adults (refer to articles with DOI: https://doi.org/10.3390/ijerph182010872).

Response:

Thank you for your comments. We have added the gap in the literature accordingly in the Introduction section, Page 4, L7-10 “However, current literature is predominately examined vaccination acceptance in the general population  or levels of knowledge concerning COVID-19 in older adults . The studies that determine the resistance and hesitancy of older population to be vaccinated are relatively limited ”.

In addition, we have mentioned knowledge gap in the original manuscript in the Introduction section, i.e., few studies have been conducted in Hong Kong to examine the factors associated with COVID-19 vaccine resistance and hesitancy, particularly those who live alone or only with an older partner, Page 5, Lines 5-6.

Methods: The survey was conducted, in part, using non-standard questions. The use of an unreliable instrument is a serious and irreversible limitation. The fact that a similar questions have been used in previous surveys is not sufficient. A validation process must be performed to evaluate the added questions to standard questionnaires. At least an evaluation of the reliability must be performed. Was a pilot study performed?

Response:

Thank you for your valuable comments and suggestions. In our study, the questionnaire was developed based on our previous studies involving vaccination status . The survey was conducted with a part of non-standard questions, i.e., self-made questions. However, the questions were developed by a group of epidemiologists, doctors, nurses and social workers. Several rounds of discussions were made until consensus was reached. To ensure the clarity and suitableness of the questions and answers, the questionnaire was also pilot tested in several elderly centres on more than 30 older adults, and minor revisions were further made. In the current study, the reliability was assessed by Cronbach’s alpha (internal consistency coefficient), with an alpha value of 0.729, showing an acceptable level of internal reliability. We have added it in the Method section accordingly, Page 6, Lines 23-24 and Page 7, Lines 1-5.

The questionnaire was developed based on our previous studies involving older adults and vaccination status . The survey included some non-standard questions, which were developed by a group of epidemiologists, doctors, nurses and social workers. To further improve the clarity and suitableness of our survey questions and answers, the questionnaire was pilot tested in several elderly centres among more than 30 older adults, and minor revisions were further made. In the current study, the reliability was evaluated by Cronbach’s alpha (internal consistency coefficient), with an alpha value of 0.729, showing an acceptable level of reliability.

The enrolment procedure must be specified. How did the authors choose the way to select the sample? And how to perform a direct interview or an online questionnaire. This can represent a great bias origin. How did they avoid the selection bias? The authors do not report a minimum sample size. What is the reference population? How large is it? Without the numerical identification of the reference population is not clear the validity of the study. A non-representative sample is by its self a non-sense-survey.

Response:

Thanks for your valuable comments. In Hong Kong, community elderly centres are mainly the first point of contact for social activities and services, to detect and address the various needs of the elderly. Social workers in the elderly centres are familiar with older people living in their community. In general, the elderly community centres would have an initial screening for older adults via phone and face-to-face interview according to their community records and referrals of their community network. Those who were eligible would then be invited to participate in our study for health assessments. All the assessments were conducted by trained social workers and volunteers, either using a paper-form questionnaire or an online questionnaire built up in the RedCap platform. But older adults themselves would not fill the questionnaire themselves, as it would be difficult in a sample of older adults with a mean age of 80 years old who had less literacy on IT techniques. As there is no data on the population who live alone or only with an older partner in community in Hong Kong, the study is not able to have such a reference population. Indeed, there would be a risk of a selection bias in our study; deprived individuals would be more likely to be invited in the study. However, in our study we mainly looked into the factors associated with vaccine resistance and hesitancy, rather than prevalence. For the minimal sample size, typically it would need about 15 samples for one variable for regression model . The current sample size (N=2109) is powered enough for the study with about 20 variables in the models. Additionally, due to the collaborating elderly community centres covering 13 out of 18 different districts with different socio-economic status in Hong Kong, the results could still provide important insights to the real situations.

To avoid confusions, the manuscript has been revised:  

Participants were recruited from their existing members and community social networks, and assessed by seven collaborating non-governmental organizations (NGOs)”, Page 5, Lines 21-22;

“The survey data were collected by either using paper-based or web-based questionnaires built in RedCap (Research Electronic Data Capture) , which is a secure and web-based data capture tools with local host at The Chinese University of Hong Kong. All collected data were entered into the RedCap platform by trained community social worker and volunteers.”, Page 6, Lines 13-17;   

“As it would need 15 samples for one variable for regression model , the minimal required sample size would be 300 with about 20 variables in the models.”, Page 8, Lines 17-19.

Selection bias has been included in our original manuscript as the first limitation.

The study had several limitations. First, this study might have not included a representative sample in community, thus, it might produce selection bias and the generalizability of our findings may be limited.”, Page 13, Lines 22-23.  

Statistical analysis: I suggest to insert a measure of the magnitude of the effect for the comparisons. Please consider to include effect sizes.

Response:

Thank you for your suggestions, we have already included effect sizes (i.e., odds ratios and their 95% confidence internals calculated by logistic regression models) to estimate the degree of association between tow binary variables in our study (See Tables 2 and 3). For example, in Table 3, compared to living with an older partner, living alone was associated with a 32% increase in vaccine resistance and hesitancy (fully adjusted odds ratio of 1.32, 95% CI 1.08-1.61), demonstrating our participants who lived alone were more likely to be resistant and hesitant to be vaccinated.

Discussion: I also suggest expanding. Emphasize the contribution of the study to the literature. The discussion must be updated with the comparison and discussion regarding knowledge in this group of population (see the above mentioned reference). The Authors should add more practical recommendations for the reader, based on their findings. Also, the section of limitations and future search is also very short, the Authors could elaborate on that.

Response:

Thank you for your valuable suggestions. In our manuscript, we have included the comparison and discussion regarding vaccine resistance and hesitancy in older adults. For example, in the reasons why participants were hesitated and resistant to be vaccinated, we compared our results with previous qualitative studies that conducted in Hong Kong and in Germany , Page 11, Line 2. Regarding the associated factors, health status was one of the main factor associated with vaccine hesitancy, which was consistent with a recent study conducted in Shanghai , Page 11, Lines 8-9.

Moreover, we have emphasized the contribution of the study to the literature and added one more Implication section to demonstrate practical recommendations for the readers, such as policy makers, and healthcare workers.

Page 13, Lines 18-20, “The findings have filled some knowledge gaps about vaccine resistance and hesitancy among vulnerable populations. The results will help policy makers, practitioners and researchers improve vaccination rates.”

Page 14, Lines 14-23, and Page 15, Lines 1-3. “Our findings highlighted the low vaccination rate among older adults who lived alone or only with an older partner. Concerns on health status and vaccine side effects were the main reasons why older people were resistant and hesitant to be vaccinated. Future health education/promotion on limited contradictions and little side effects of vaccine are important to achieve high levels of vaccination coverage in this population. It is vital to connect with community health service centres to sharing clear and timely information about COVID-19 vaccines (e.g., safety and efficacy) and vaccination sites and addressing misinformation. Healthcare professionals and workers may be encouraged to help answer enquiries from the elderly to strengthen their confidence in vaccination. Tailored messages and content, both online and offline (e.g., leaflets), that are relative to older adults are required to increase their engagement. Follow-up services (e.g., phone calls) after getting vaccinated would also help relieve their vaccine anxiety.

Furthermore, we have improved our first Limitation with future research suggestion accordingly, “However, due to the collaborating elderly community centres covering 13 out of 18 different districts with different socio-economic status in Hong Kong, the results could still provide important insights to the real situations. More studies with a random sample of older adults (e.g., household surveys) who lived alone or only with an older partner are required to examine the factors associated with vaccine hesitancy and resistance.” Page 13, Lines 22 and Page 14, Lines 1-4,

Round 2

Reviewer 2 Report

The authors tried to improve the manuscript but main concerns remain. Overall, the used tool and it validation. Second the sampling method nont standardized and not reliable. Third, the reference population and its minimum sample size. References are scarce and inadeguate. The replies of the Authors are not acceptable for a journal such Vaccines:

 "the study is not able to have such a reference population" this is the first step for a well conduced study 

"For the minimal sample size, typically it would need about 15 samples for one variable for regression model" typically?? the minimum sample size must be stated at the beginning of the study

A non-representative sample is by its self a non-sense-survey.

Author Response

Response:

We appreciate the reviewer for the positive and very valuable comments on our paper. Vaccinating older adults, particularly those who live alone or only with an older partner, is one of the public health priorities across the world. Understanding reasons why older adults were resistant and hesitant and associated factors would be of significance to achieve high levels of vaccination coverage. We have revised and improved our manuscripts accordingly. The one-to-one responses are shown below.

Title: it can be improved, highlight the object of the study: place, time and person.

Response:

Thank you. We have revised the title accordingly as “Vaccine resistance and hesitancy among older adults who live alone or only with an older partner in community in the early stage of the fifth wave of COVID-19 in Hong Kong”, Page 1, Lines 1-3.

Introduction: The authors should make clearer what is the gap in the literature that is filled with this study. The authors must better frame their study within the vast body of literature that addressed the issue of acceptance of the vaccination in the adult population and their level of knowledge concerning COVID-19 in older adults (refer to articles with DOI: https://doi.org/10.3390/ijerph182010872).

Response:

Thank you for your comments. We have added the gap in the literature accordingly in the Introduction section, Page 4, L7-10 “However, current literature is predominately examined vaccination acceptance in the general population  or levels of knowledge concerning COVID-19 in older adults . The studies that determine the resistance and hesitancy of older population to be vaccinated are relatively limited ”.

In addition, we have mentioned knowledge gap in the original manuscript in the Introduction section, i.e., few studies have been conducted in Hong Kong to examine the factors associated with COVID-19 vaccine resistance and hesitancy, particularly those who live alone or only with an older partner, Page 5, Lines 5-6.

Methods: The survey was conducted, in part, using non-standard questions. The use of an unreliable instrument is a serious and irreversible limitation. The fact that a similar questions have been used in previous surveys is not sufficient. A validation process must be performed to evaluate the added questions to standard questionnaires. At least an evaluation of the reliability must be performed. Was a pilot study performed?

Response:

Thank you for your valuable comments and suggestions. In our study, the questionnaire was developed based on our previous studies involving vaccination status . The survey was conducted with a part of non-standard questions, i.e., self-made questions. However, the questions were developed by a group of epidemiologists, doctors, nurses and social workers. Several rounds of discussions were made until consensus was reached. To ensure the clarity and suitableness of the questions and answers, the questionnaire was also pilot tested in several elderly centres on more than 30 older adults, and minor revisions were further made. In the current study, the reliability was assessed by Cronbach’s alpha (internal consistency coefficient), with an alpha value of 0.729, showing an acceptable level of internal reliability. We have added it in the Method section accordingly, Page 6, Lines 23-24 and Page 7, Lines 1-5.

The questionnaire was developed based on our previous studies involving older adults and vaccination status . The survey included some non-standard questions, which were developed by a group of epidemiologists, doctors, nurses and social workers. To further improve the clarity and suitableness of our survey questions and answers, the questionnaire was pilot tested in several elderly centres among more than 30 older adults, and minor revisions were further made. In the current study, the reliability was evaluated by Cronbach’s alpha (internal consistency coefficient), with an alpha value of 0.729, showing an acceptable level of reliability.

The enrolment procedure must be specified. How did the authors choose the way to select the sample? And how to perform a direct interview or an online questionnaire. This can represent a great bias origin. How did they avoid the selection bias? The authors do not report a minimum sample size. What is the reference population? How large is it? Without the numerical identification of the reference population is not clear the validity of the study. A non-representative sample is by its self a non-sense-survey.

Response:

Thanks for your valuable comments. In Hong Kong, community elderly centres are mainly the first point of contact for social activities and services, to detect and address the various needs of the elderly. Social workers in the elderly centres are familiar with older people living in their community. In general, the elderly community centres would have an initial screening for older adults via phone and face-to-face interview according to their community records and referrals of their community network. Those who were eligible would then be invited to participate in our study for health assessments. All the assessments were conducted by trained social workers and volunteers, either using a paper-form questionnaire or an online questionnaire built up in the RedCap platform. But older adults themselves would not fill the questionnaire themselves, as it would be difficult in a sample of older adults with a mean age of 80 years old who had less literacy on IT techniques. As there is no data on the population who live alone or only with an older partner in community in Hong Kong, the study is not able to have such a reference population. Indeed, there would be a risk of a selection bias in our study; deprived individuals would be more likely to be invited in the study. However, in our study we mainly looked into the factors associated with vaccine resistance and hesitancy, rather than prevalence. For the minimal sample size, typically it would need about 15 samples for one variable for regression model . The current sample size (N=2109) is powered enough for the study with about 20 variables in the models. Additionally, due to the collaborating elderly community centres covering 13 out of 18 different districts with different socio-economic status in Hong Kong, the results could still provide important insights to the real situations.

To avoid confusions, the manuscript has been revised:  

Participants were recruited from their existing members and community social networks, and assessed by seven collaborating non-governmental organizations (NGOs)”, Page 5, Lines 21-22;

“The survey data were collected by either using paper-based or web-based questionnaires built in RedCap (Research Electronic Data Capture) , which is a secure and web-based data capture tools with local host at The Chinese University of Hong Kong. All collected data were entered into the RedCap platform by trained community social worker and volunteers.”, Page 6, Lines 13-17;   

“As it would need 15 samples for one variable for regression model , the minimal required sample size would be 300 with about 20 variables in the models.”, Page 8, Lines 17-19.

Selection bias has been included in our original manuscript as the first limitation.

The study had several limitations. First, this study might have not included a representative sample in community, thus, it might produce selection bias and the generalizability of our findings may be limited.”, Page 13, Lines 22-23.  

Statistical analysis: I suggest to insert a measure of the magnitude of the effect for the comparisons. Please consider to include effect sizes.

Response:

Thank you for your suggestions, we have already included effect sizes (i.e., odds ratios and their 95% confidence internals calculated by logistic regression models) to estimate the degree of association between tow binary variables in our study (See Tables 2 and 3). For example, in Table 3, compared to living with an older partner, living alone was associated with a 32% increase in vaccine resistance and hesitancy (fully adjusted odds ratio of 1.32, 95% CI 1.08-1.61), demonstrating our participants who lived alone were more likely to be resistant and hesitant to be vaccinated.

Discussion: I also suggest expanding. Emphasize the contribution of the study to the literature. The discussion must be updated with the comparison and discussion regarding knowledge in this group of population (see the above mentioned reference). The Authors should add more practical recommendations for the reader, based on their findings. Also, the section of limitations and future search is also very short, the Authors could elaborate on that.

Response:

Thank you for your valuable suggestions. In our manuscript, we have included the comparison and discussion regarding vaccine resistance and hesitancy in older adults. For example, in the reasons why participants were hesitated and resistant to be vaccinated, we compared our results with previous qualitative studies that conducted in Hong Kong and in Germany , Page 11, Line 2. Regarding the associated factors, health status was one of the main factor associated with vaccine hesitancy, which was consistent with a recent study conducted in Shanghai , Page 11, Lines 8-9.

Moreover, we have emphasized the contribution of the study to the literature and added one more Implication section to demonstrate practical recommendations for the readers, such as policy makers, and healthcare workers.

Page 13, Lines 18-20, “The findings have filled some knowledge gaps about vaccine resistance and hesitancy among vulnerable populations. The results will help policy makers, practitioners and researchers improve vaccination rates.”

Page 14, Lines 14-23, and Page 15, Lines 1-3. “Our findings highlighted the low vaccination rate among older adults who lived alone or only with an older partner. Concerns on health status and vaccine side effects were the main reasons why older people were resistant and hesitant to be vaccinated. Future health education/promotion on limited contradictions and little side effects of vaccine are important to achieve high levels of vaccination coverage in this population. It is vital to connect with community health service centres to sharing clear and timely information about COVID-19 vaccines (e.g., safety and efficacy) and vaccination sites and addressing misinformation. Healthcare professionals and workers may be encouraged to help answer enquiries from the elderly to strengthen their confidence in vaccination. Tailored messages and content, both online and offline (e.g., leaflets), that are relative to older adults are required to increase their engagement. Follow-up services (e.g., phone calls) after getting vaccinated would also help relieve their vaccine anxiety.

Furthermore, we have improved our first Limitation with future research suggestion accordingly, “However, due to the collaborating elderly community centres covering 13 out of 18 different districts with different socio-economic status in Hong Kong, the results could still provide important insights to the real situations. More studies with a random sample of older adults (e.g., household surveys) who lived alone or only with an older partner are required to examine the factors associated with vaccine hesitancy and resistance.” Page 13, Lines 22 and Page 14, Lines 1-4, 

Our findings highlighted the low vaccination rate among older adults who lived alone or only with an older partner. Concerns on health status and vaccine side effects were the main reasons why older people were resistant and hesitant to be vaccinated. Future health education/promotion on limited contradictions and little side effects of vaccine are important to achieve high levels of vaccination coverage in this population. It is vital to connect with community health service centres to share clear and timely information about COVID-19 vaccines (e.g., safety and efficacy) and vaccination sites and address misinformation. Healthcare professionals and workers may be encouraged to help answer enquiries from the elderly to strengthen their confidence in vaccination. Tailored messages and content, both online and offline (e.g., leaflets), that are relative to older adults are required to increase their engagement. Follow-up services (e.g., phone calls) after getting vaccinated would also help relieve their vaccine anxiety.”    

Round 3

Reviewer 2 Report

I confirm my previouse evaluation
